# ETHIAD: A novel explainable model for detecting illicit accounts on Ethereum

**Jiarong Lu** [1]*, **Bin Liao**[2], **Yi Liu**[3], **Kutorzi Edwin Yao**[4]

1 College of Statistics and Data Science, Xinjiang University of Finance and Economics, Urumqi, PR China, 2 College of Big Data Statistics, Guizhou University of Finance and Economics, Guiyang, PR China, 3 School of Public Health, Xinjiang Medical University, Urumqi, PR China, 4 School of Mathematics and Institute for Financial Studies, Shandong University, Jinan, PR China

* ljrong1203@126.com

## Abstract

Ethereum has become a significant trading platform for financial activities such as Dapps, ICOs, and DeFi. However, it has also become a hub for criminal activities such as fraud, money laundering, and illicit fundraising. The construction of fraud detection models employing machine learning techniques is currently a mainstream research direction. Nevertheless, existing studies face significant challenges, including class imbalance in data samples and a lack of model interpretability. In this content, this work proposes a novel explainable model for Ethereum illicit account detection, ETHIAD (Ethereum Illicit Account Detection). Firstly, we pre-process the dataset by ADASYN oversampling and Lasso feature selection, etc., to more efficiently achieve feature modeling of transaction structures. Then, the ETHIAD model is trained using the XGboost algorithm, with an accuracy, precision, recall, F1 score, and AUC value of 99.70%, 99.51%, 99.02%, 99.26%, and 99.45%, respectively, the model outperforms the existing SOTA model by 0.05%−1.1%. Finally, we introduce SHAP framework to analyze the key influencing factors of illicit accounts from multiple perspectives, and the conclusions strongly enhance the explainability of the model.

## 1. Introduction

Blockchain is a distributed ledger technology that is decentralized, anonymous, and untameable, it's become increasingly popular since Bitcoin was created in 2008 [1,2]. Today, blockchain technology is being used for various things beyond cryptocurrency trading. such as supply chain management, healthcare, software engineering, etc [3–7]. Obviously, one of the most important applications of blockchain technology is cryptocurrency. Ethereum is an example of a successful cryptocurrency created in 2014 by Vitalik Buterin through an Initial Coin Offering (ICO), especially, Ethereum combines blockchain technology and smart contract technology [8], and gradually

**Data availability statement:** Datasets and source code are available from the GitHub repository: https://github.com/lujiarong1203/ETHIAD.

**Funding:** This work was supported by Research on Digital-Intelligence-Driven Technology Innovation Models and Influencing Factors in Xinjiang's Industrial Enterprises (橫 20250035), Guizhou Provincial Basic Research Program (Natural Science) MS [2025] 226. The funders had no role in study design, data collection and analysis, decision to publish, or preparation of the manuscript.

**Competing interests:** The authors have declared that no competing interests exist.

develops into a significant trading platform for decentralized applications, ICOs, DeFi (decentralized finance), and other financial activities [9].

However, the decentralization and anonymity of blockchain technology have made it a breeding ground for criminal activities such as money laundering, phishing, illicit fundraising, and Ponzi schemes. An example is the attack on The DAO [10], the largest crowdfunding project in the blockchain industry, in 2016, The hacker used the address 0x F35e... a77D to transfer 3.6 million ETH, worth more than $60 million at the time. This is just one example of the many recent fraud cases that have rocked the cryptocurrency community. As of December 2020, over 358 major blockchain security incidents have led to over $14.2 billion in economic losses [11]. According to The Chainalysis 2023 Crypto Crime Report, USD 39.6 billion worth of cryptoassets were received by identified illicit addresses, accounting for 0.42% of total on-chain transaction volume—a significant increase from the previous year's USD 23.2 billion [12]. Undoubtedly, blockchain transaction security issues have seriously hindered the development of technology and financial innovation. Therefore, accurately detecting illicit accounts in blockchain transaction networks, identifying the salient characteristics of illicit accounts, and providing investors with reliable risk warning services are urgent problems for financial regulators and service organizations.

Constructing fraud detection models from Ethereum transaction data has proven an effective approach to mitigating blockchain security risks, as it allows efficient utilization of on-chain information for the rapid identification of abnormal behaviors. However, existing studies still encounter several inherent challenges. First, the extreme class imbalance between illicit and normal accounts biases model learning toward the majority class, reducing sensitivity to high-risk entities. Second, feature redundancy and noise within high-dimensional transaction data hinder model stability and generalization. Third, most detection models lack interpretability, making it difficult to understand the underlying decision logic and thereby limiting their applicability in regulatory and forensic contexts.

In response to these challenges, this study aims to develop a robust and interpretable detection framework capable of accurately identifying illicit Ethereum accounts from large-scale transactional data. To this end, we propose ETHIAD, a novel model that systematically mitigates data imbalance, suppresses redundant features, and enhances interpretability through an integrated pipeline of adaptive resampling, sparse feature selection, and explainable learning.

The main contributions of this study are as follows:

- **Adaptive feature modeling**. We establish an efficient feature engineering process for Ethereum transaction networks by combining One-Hot encoding with ADASYN oversampling and Lasso regularization, enabling balanced and noise-resistant feature representation.

- **Robust and high-performing detection model.** We construct the ETHIAD framework based on XGBoost, which achieves state-of-the-art detection accuracy while maintaining strong generalization on unseen data.

- **Interpretability through attribution analysis**. We incorporate the SHAP framework to quantify feature contributions, uncovering key behavioral signatures of illicit accounts and enhancing the model's transparency and credibility for real-world monitoring applications..

## 2. Literature review

The Ethereum blockchain account's identity information is encrypted to ensure strict security. However, its data structure inherently records and publicly exposes all transaction details, including the transaction hash (TxHash), sender and receiver addresses, transferred amount, and timestamp. This transparent and immutable ledger enables every transaction to be verified and traced across the entire network. Such transparency allows researchers to reconstruct complete transaction flows between accounts and to extract behavioral and structural indicators—such as transaction frequency, temporal patterns, and ERC20 token transfer dynamics—that are crucial for distinguishing normal and illicit accounts. In this study, these traceable and quantifiable characteristics are leveraged as core input variables for detecting illicit accounts, providing a robust foundation for interpretable fraud detection modeling.

Building upon this transparent transaction environment, recent studies on blockchain-based fraud detection have generally focused on three main directions: (a) constructing fraud detection models based on manually engineered transaction features; (b) employing graph embedding techniques to automatically learn node (address) representations for downstream classification [13–16]; and (c) utilizing Graph Neural Networks (GNNs) to build end-to-end fraud detection models [17–19]. The first approach provides simplicity, efficiency, and high interpretability but depends heavily on feature quality; the second automatically mines node information from graph structures but increases computational complexity and reduces transparency; and the third achieves end-to-end optimization but requires high-performance hardware and still suffers from poor interpretability.

This paper focuses on the study of building fraud detection models based on basic transaction data. As a pioneering and contributing level study, Farrugia et al. [20] collected 3,000 normal account addresses from blocks 3,800,000–3,805,000 and 2,179 illicit account addresses flagged by the Ethereum community. They obtained Ethereum's basic transaction data using the Ethereum API and developed an illicit account detection model based on the XGboost algorithm. The accuracy rate of the model was 96.3%, and the AUC value was 99.4%. The feature extraction tools developed by Farrugia et al. [20] have been widely adopted by subsequent studies. Aziz et al. [21] proposed an ethereum fraudulent transaction detection method based on several machine learning algorithms (e.g., LightGBM, Random Forest, MLP) and the LightGBM algorithm had the highest accuracy of 98.60% in a specific dataset scenario. IIbrahim et al. [22] proposed a fraud detection model using three different machine learning algorithms (Decision Trees, Random Forest, KNN) and used correlation coefficients to select the most effective features and constructed a new dataset using only six features, and the experimental results showed that KNN's precision, recall, and F1 score reached 97.4%, 97.5%, and 97.4% respectively. Chen et al. [23] propose a graph-based cascade feature extraction method based on transaction records and a lightGBM-based Dual-sampling Ensemble algorithm to build the identification model. Extensive experiments shown that the proposed algorithm can effectively identify phishing scams. Wen et al. [24] proposed a phishing detection framework based on feature learning and a phishing hiding framework based on inserting transaction records, respectively. The experimental results show the effectiveness of the constructed phishing detection framework and the superiority of the phishing hiding strategy, in which the precision rate, recall rate, F1 value, and AUC value of the model constructed using the Adaboost algorithm reach 87.5%, 81.5%, 84%, and 92.76%, respectively. Kabla et al. [25] proposes a detection mechanism called Ethereum Phishing Scam Detection (Eth-PSD) that attempts to detect phishing scam-related transactions using a novel machine learning-based approach, Eth-PSD tackles some of the limitations in the existing works, such as the use of imbalanced datasets, complex feature engineering, and lower detection accuracy, the experimental results show that Eth-PSD can effectively detect phishing scams on Ethernet with a detection accuracy of 98.11%.

In summary, although existing studies have achieved considerable progress in Ethereum fraud detection, several critical challenges remain unresolved. (1) The datasets used in prior research are typically highly imbalanced, causing models to insufficiently learn the behavioral characteristics of illicit accounts and consequently biasing the detection outcomes. (2) The high dimensionality and sparsity of Ethereum transaction data easily trigger the "curse of dimensionality," leading to overfitting and increased computational complexity during model training. (3) Existing approaches generally lack interpretability, making it difficult to identify the intrinsic mechanisms and behavioral indicators underlying fraudulent activities.

To address these challenges, this study aims to develop a robust and explainable detection framework, termed ETHIAD (Ethereum Illicit Account Detection), that achieves both high performance and interpretability. Specifically, we employ One-Hot encoding, ADASYN oversampling, and Lasso-based feature selection to construct a more balanced and compact feature representation, thereby enhancing the model's learning efficiency and robustness. The ETHIAD framework, built upon the XGBoost algorithm, achieves state-of-the-art detection accuracy while maintaining stable generalization to unseen data. Furthermore, by integrating the SHAP interpretability framework, we systematically quantify and visualize the key contributing factors of illicit accounts from multiple perspectives, thereby improving the transparency and practical applicability of the model in real-world financial monitoring.

## 3. Model framework and methodology

This chapter has three main sections. Section 3.1 talks about the Problem Setting and the ETHIAD Model Building Framework. Section 3.2 provides an overview of the principles of the base model XGboost algorithm, and Section 3.3 briefly describes the SHAP framework. Additionally, **Table 1** lists the symbolic descriptions of the mathematical formulas used in this article.

**Table 1. Definition of mathematical formula symbols.**

| Number | Symbol | Meaning |
|---|---|---|
| 1 | $D$ | The raw Ethereum basic transaction dataset |
| 2 | $D'$ | Pre-processed dataset |
| 3 | $X^M$ | Feature set space, where M is the number of features |
| 4 | $x_i^j$ | Input sample instance: the value of the $j$-th feature of the $i$-th sample |
| 5 | $Y$ | The target Space |
| 6 | $y_i$ | The true value of the $i$-th sample |
| 7 | $\hat{y}_i$ | The predicted value of the $i$-th sample |
| 8 | $T$ | The training set |
| 9 | $ObF^{(t)}$ | The objective function of the $t$-th tree for training |
| 10 | $l$ | Loss function |
| 11 | $\Omega$ | Regularization function |
| 12 | $\gamma$ | Penalized regular terms for trees |
| 13 | $\lambda$ | Penalized regular terms for weights |
| 14 | $\omega_i$ | Output scores for each tree leaf node |
| 15 | $G_i$ | The first-order partial derivative of the loss function for the $i$-th sample |
| 16 | $H_i$ | The second-order partial derivative of the loss function for the $i$-th sample |
| 17 | $y_{base}$ | Baseline value for model prediction |
| 18 | $\varphi$ | The SHAP value of the feature |

## 3.1. Problem definition and ETHIAD framework

This study aims to accurately detect illicit Ethereum accounts and identify the key behavioral features that distinguish them from legitimate accounts. Formally, let the original Ethereum transaction dataset be $D$, and the feature space be $X^M$, where $M$ is the number of features, and the label space be $Y$. Let the sample instances of the input model be $x_i^j = (x_i^1, x_i^2, \ldots, x_i^M)$, and the output of the model prediction be $\hat{y} = (\hat{y}_1, \hat{y}_2, \ldots, \hat{y}_n)$, where $i = 1, 2, \ldots, N$, and $N$ is the sample capacity. Therefore, this study aims to learn a function $\mathcal{F}$, denoted as:

$$\mathcal{F}: X^M \to Y \quad \text{or} \quad \hat{y} = \mathcal{F}(x_i^j)$$

(1)

As shown in **Fig 1**, the work of this paper is mainly divided into two parts: the first part is feature modeling, which includes primarily missing value filling, One-Hot coding, sample sampling, and feature selection, etc.; the second part is modeling

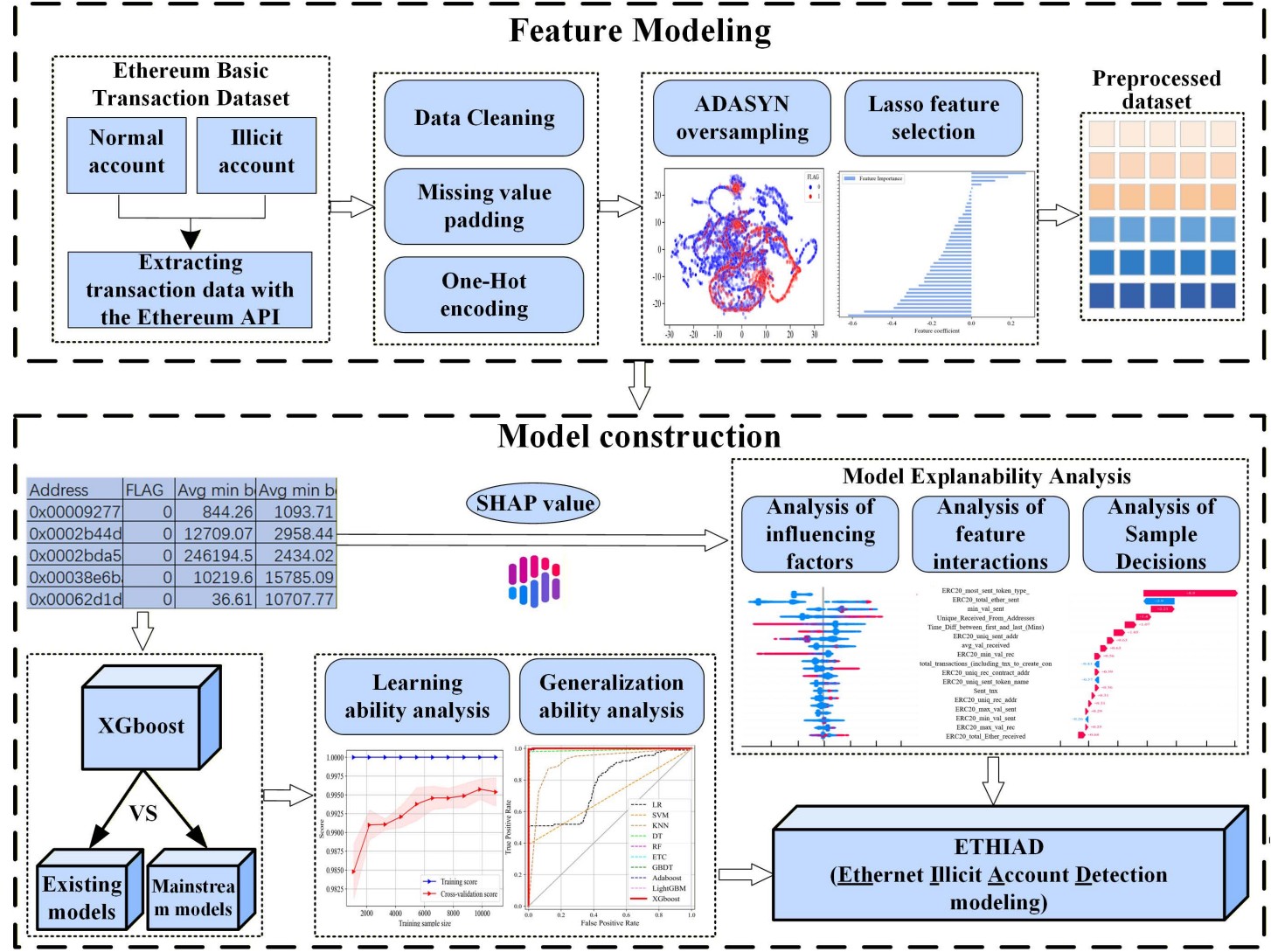

**Fig 1. Modeling flow of ETHIAD.**

and interpretability analysis, which mainly comprises model training, comparing with the existing work, hyper-parameter tuning, generalization ability analysis and illicit account influencing factor analysis.

The construction process of ETHIAD is presented in Algorithm 1. We begin with the raw Ethereum transaction dataset, the list of parameters to adjust, and four models used to analyze the importance of features. The output is a trained model, the optimal hyperparameter combination, and the primary factors affecting illicit accounts. The first step removes useless features and pads missing values, and One-Hot encodes the data. Next, we split the dataset into a 90% training set and a 10% test set to avoid data snooping. In rows 3 and 4, only ADASYN oversampling and Lasso feature selection are performed on the training set. This completes feature modeling for the Ethereum transaction network. In line 5, we select the XGboost algorithm to train the initial model on the training set, and in lines 6–8, we tune the hyperparameters of the initial model. The model's generalization ability is verified on the test set in lines 9–11. In lines 12–16, we compare feature importance calculated by multiple models to derive the primary factors affecting illicit accounts on Ethereum. The SHAP framework is introduced in line 17 to comprehensively analyze the primary factors and obtain the most salient features of illegal accounts. Based on this analysis, we derive the interpretable Ethereum illicit account detection model ETHIAD in line 18 and output all results in line 19.

```
Algorithm 1: Model building algorithms for ETHIAD
INPUT: Raw Ethereum transaction data: D
  Panding_Params_List: {'n_estimators': range(0.1, 1), step=1; 'n_estimators': range(90, 160),
  step=10; 'max_depth': range(1, 10), step=1; 'subsample': range(0.1, 1), step=1; 'colsample_
  bytree': range(0.1, 1), step=1.
  ModelList: [Random Forest, LightGBM, XGboost, SHAP].

OUTPUT: ETHIAD, The Best_parameters, Main influences of illicit accounts.
1 D' = (X^M, Y) ← process (D), Including the process of eliminating meaningless features, filling in
  missing values, One-Hot encoding, etc.
2 Training set, Test set←Split D' into 90% training set and 10% test set
3 T ← ADASYN over-sampling (Training set)
4 T ← Lasso feature selection (T)
5 Initial_model←XGboost(Default parameters).fit (T)
6 Parameter_list ← {Panding_Params_List}
7 Grid_search←GridSearchCV (Initial_model, Parameter_list, scoring="accuracy", cv=10)
8 The Best_parameters←Grid_search.fit(T)
9 The Best Model←Initial_model(The Best_parameters)
10 Evaluate the model's generalization ability on the test set
11 Performance of The Best Model
12 for model in ModelList do
13    model.fit(T)
14    feature importance←model.feature_importances_
15 end for
16 Main features←Compare the feature importance rankings of the 4 models
17 Main influences of illicit accounts←SHAP(Main features)
18 ETHIAD←The Best Model+Main influences of illicit accounts
19 return ETHIAD, The Best_parameters, Main influences of illicit accounts.
```

## 3.2. XGboost algorithm

ETHIAD selects XGBoost [26] as the core learner due to its ability to effectively model the high-dimensional, sparse, and nonlinear characteristics of Ethereum transaction data, which are challenging for simpler algorithms such as Decision Trees, KNN, or AdaBoost. XGBoost's gradient boosting framework, combined with regularization and multi-threaded computation, enables robust learning while mitigating overfitting and ensuring computational efficiency

[27–29]. This makes it particularly suitable for accurately detecting illicit accounts and capturing subtle behavioral patterns. In the following, we provide a concise theoretical overview of the XGBoost algorithm and its application within ETHIAD:

(1) Let the Ethereum transaction dataset $D = \{(x_i^j, y_i)\}$, $i = 1, 2, \cdots, N$, $j = 1, 2, \cdots, M$, and the results of $K$ iterations are used as the outputs of the boosting tree $f$. Then the predicted price $\hat{y}_i$ for the $i$-th account sample $x_i^j$ is denoted as Eq. (2).

$$\hat{y}_i = \sum_{k=1}^{K} f_k(x_i^j) \tag{2}$$

(2) The loss function during the training of the illicit account detection model is shown in Eq. (3)

$$Obj = \sum_i l(y_i, \hat{y}_i) + \sum_k \Omega(f_k) \tag{3}$$

$$\Omega(f_k) = \gamma T + \frac{1}{2}\lambda \sum_{j=1}^{M} \omega_j^2 \tag{4}$$

Where, $\sum_i l(y_i, \hat{y}_i)$ represents the loss function, $\sum_k \Omega(f_k)$ represents the regularization term, $\gamma$ is the penalty term of the tree, and $\omega_i$ is the output score of the leaf node of each tree.

(3) A gradient boosting strategy is used in the model training process, i.e., the existing model is retained and a new regression tree is added to the model each time. Let the prediction result of the $i$-th sample in the $t$-th iteration be $\hat{y}_i^{(t)}$, $f_t(x_i^j)$ is the new regression tree added, and the derivation process is shown in the following.

$$\hat{y}_i^{(0)} = 0$$

$$\hat{y}_i^{(1)} = f_1(x_i^j) = \hat{y}_i^{(0)} + f_1(x_i^j)$$

$$\hat{y}_i^{(2)} = f_1(x_i^j) + f_2(x_i^j) = \hat{y}_i^{(1)} + f_2(x_i^j)$$

$$\vdots$$

$$\hat{y}_i^{(t)} = \sum_{k=1}^{t} f_k(x_i^j) = \hat{y}_i^{(t-1)} + f_t(x_i^j) \tag{5}$$

Substituting Eq. (5) into Eq. (3) yields Eq. (6).

$$Obj^{(t)} = \sum_{i=1}^{N} l(y_i, \ \hat{y}_i^{(t-1)} + f_t(x_i^j)) + \Omega(f_k) + constant \tag{6}$$

(5) A second-order Taylor expansion of the loss function is performed and a regular term is introduced as shown in Eq. (7).

$$Obj^{(t)} \cong \sum_{i=1}^{N} [g_i f_t(x_i^j) + \frac{1}{2} h_i f_t^2(x_i^j)] + \Omega(f_t) = \sum_{i=1}^{N} [g_i \omega_{q(x)} + \frac{1}{2} h_i \omega_{q(x)}^2] + \gamma T + \frac{1}{2}\lambda \sum_{j=1}^{T} \omega_j^2 = \sum_{j=1}^{M} [(\sum_{i\in I_j} g_i)\omega_j + \frac{1}{2}(\sum_{i\in I_j} h_i + \lambda)\omega_j^2] + \gamma T \tag{7}$$

where $g_i = \partial_{\hat{y}_i^{(t-1)}} l(y_i, \ \hat{y}_i^{(t-1)})$, and $h_i = \partial^2_{\hat{y}_i^{(t-1)}} l(y_i, \ \hat{y}_i^{(t-1)})$. Defining $G_i = \sum_{i\in I_j} g_i$, $H_i = \sum_{i\in I_j} h_i$, then Eq. (7) reduces to the form of Eq. (8).

$$Obj^{(t)} = \sum_{j=1}^{T} [G_i \omega_j + \frac{1}{2}(H_i + \lambda)\omega_j^2] + \gamma T \tag{8}$$

where the leaf node output score $\omega_i$ is an uncertain value, so by taking the first-order derivative of the loss function $Obj^{(t)}$ with respect to $\omega_i$, the optimal value of the output score $\omega_j^*$ for the jth leaf node can be found as shown in Eq. (9).

$$\omega_j^* = -\frac{1}{H_i + \gamma} \tag{9}$$

Substituting $\omega_j^*$ into the loss function, $Obj^{(t)}$ obtains the minimum value as shown in Eq. (10).

$$Obj^{(t)} = -\frac{1}{2} \sum_{j=1}^{T} \frac{G_i}{H_i + \gamma} + \gamma T \tag{10}$$

### 3.3. SHAP explainability framework

To better uncover the underlying behavioral factors of illicit accounts on Ethereum, this study integrates the SHapley Additive exPlanations (SHAP) framework into the analysis to overcome the limited interpretability of ensemble models such as XGBoost. Originally proposed by Lundberg and Lee [30], the SHAP framework has been widely adopted to enhance the transparency and interpretability of machine learning models [31–34]. Its core principle is to quantify the marginal contribution of each feature to an individual prediction, thereby providing a theoretically grounded explanation of model behavior.

In the context of Ethereum illicit account detection, each account is regarded as a prediction instance, and each transaction feature acts as a "participant" contributing to the final prediction outcome. The SHAP value of a given feature represents the degree and direction of its contribution to the model's output, allowing us to distinguish whether the feature increases or decreases the likelihood of an account being classified as illicit. Formally, for each account instance $x_i$, the model prediction process can be represented as:

$$\hat{y}_i = y_{base} + \sum_{j=1}^{M} \varphi\left(x_i^j\right) \tag{11}$$

where $\hat{y}_i$ denotes the model's prediction for instance $i$, $y_{base}$ is the baseline output (the average prediction across all samples), and $\phi(x_i^j)$ is the SHAP value corresponding to feature $j$. A positive SHAP value ($\phi(x_i^j) > 0$) indicates that the feature contributes positively to the prediction of an illicit account, whereas a negative value ($\phi(x_i^j) < 0$) suggests a mitigating or normalizing influence.

Unlike traditional feature importance metrics that merely quantify feature relevance at the global level, the SHAP framework offers both local and global interpretability, revealing not only *which* features are important but also *how* they influence individual predictions. In this study, SHAP analysis enables a fine-grained exploration of the behavioral characteristics of illicit accounts—such as abnormal transaction frequencies, out-degree concentrations, or excessive transfer volumes—thereby bridging the gap between model performance and behavioral interpretability in Ethereum fraud detection.

## 4. Experiments and results analysis

### 4.1. Experimental environment and evaluation metrics

The experiments were conducted on a 64-bit Win10 operating system with 8GB RAM, using Python 3.9 and scikit-learn 1.0.1 called on the Pycharm 2022.3 platform running on an Intel (R) Core i7-7500U CPU 2.7GHz.

In this paper, five classification algorithm evaluation indicators were selected to measure the detection performance of the model. These indicators include accuracy (Acc), precision (Pre), recall (Rec), F1 value (F1), and AUC value. They are defined as follows:

$$Accuracy = \frac{TP + TN}{TP + FP + FN + TN} \tag{12}$$

$$Precision = \frac{TP}{TP + FP} \tag{13}$$

$$Recall = TPR = \frac{TP}{TP + FN} \tag{14}$$

$$F1 - score = \frac{2 \times Precision \times Recall}{Precision + Recall} \tag{15}$$

The terms TP, FP, TN, and FN represent different predictions made by a model while classifying accounts as illicit or regular. TP denotes the number of samples correctly predicted as illicit accounts, while FP denotes the number of samples incorrectly predicted as illicit accounts. Similarly, TN represents the number of samples correctly predicted as regular accounts, while FN represents the number of samples incorrectly predicted as regular accounts. The AUC value indicates the area under the ROC curve, which reflects the model's performance in predicting actual positive rate (TPR) samples versus false positive rate (FPR) under different classification thresholds. A higher AUC value indicates better classification performance, with 1 being the optimal value.

### 4.2. Data description and feature modeling

In this study, we adopt the Ethereum illicit account detection dataset originally compiled by Farrugia et al. [20], which comprises 7,662 normal accounts and 2,179 illicit accounts. This dataset has been used in multiple subsequent studies,

making it a widely recognized benchmark for Ethereum fraud detection. The dataset contains detailed transaction-level features, including the average time between outgoing transactions, the time interval between first and last transactions, total numbers of sent and received transactions, and ERC-20 token transfer details. Such features are directly relevant to our research objective of detecting illicit accounts and analyzing their behavioral patterns. Following standard preprocessing, features with zero variance across samples are removed, yielding a final dataset of 40 feature variables and one target label (see **Table A1** for details).

To better realize the feature modeling of Ethereum transaction information and improve the prediction ability and robustness of the model, this study divides the original dataset into a 90% training set and a 10% test set. Then, it performs missing value filling and One-Hot encoding on the complete dataset, and feature selection and sample sampling preprocessing are carried out on the training set.

(1) Missing value padding

To ensure the completeness and reliability of the dataset before model construction, missing value imputation is the first step of feature engineering. ERC-20 is a token standard on the Ethereum blockchain. Tokens following ERC-20 can be compatible with the Ethereum network, making it a widely used token standard for ICO (Initial Coin Offering) and crowdfunding projects. Therefore, ERC-20 features are crucial for identifying illicit accounts. This study used K-Nearest Neighbors Imputation to estimate the missing values in ERC-20 features. This method considers the similarity between samples to retain the structure and relationship of the original data as much as possible. It provides a more reliable basis for subsequent data analysis and model construction.

(2) One-Hot Encoding

Since machine learning algorithms cannot directly process categorical strings, we transform token-type variables into numerical vectors to facilitate model learning. The features ERC20_most_sent_token_type and ERC20_most_rec_token_type are string features, which are the types of tokens that the account sends and receives the most tokens through ERC-20 transactions, respectively. These features can help identify accounts involved in fraud, money laundering, or other illicit activities. The word cloud of the feature ERC20_most_sent_token_type shows that the most frequently sent token types via ERC20 transactions are Blockwell, EOS, Omise, and others. However, most accounts do not have information regarding ERC20 token transactions. One-Hot encoding converts these string features into binary sparse features, which expands the feature dimension to 812 dimensions. The model's performance is compared before and after One-Hot encoding in **Table 2**, and it is observed that the model's performance has improved after processing.

(3) Sample Sampling

To prevent bias caused by imbalanced classes and to enhance the model's ability to identify illicit accounts, the data distribution must be balanced before training. Category imbalance causes the model to be biased towards the majority of category samples during training, which can reduce the generalization ability of the model [35]. Furthermore, ambiguous boundaries may cause the classifier to be more sensitive to noise or abnormal samples, which weakens the model's robustness. Several sampling methods are available to address this issue, including oversampling of minority samples, undersampling of majority samples, and mixed sampling. For instance, as shown in **Fig 2(a)**, the proportion of illicit accounts in the Ethereum transaction dataset is 22.1%. To enhance the model's ability to classify illicit accounts and

**Table 2. Comparison of model performance before and after One-Hot coding.**

|  | Feature Dimension | Acc/ % | Pre/ % | Rec/ % | F1/ % | AUC/ % |
|---|---|---|---|---|---|---|
| Before encoding | 38 | 98.68 | 96.59 | 97.06 | 96.82 | 98.08 |
| After encoding | 812 | 99.49 | 99.50 | 98.04 | 98.77 | 98.96 |

to better learn their distinguishing features, we use different sampling algorithms SMOTE, ADASYN, TomekLink, and SMOTE+TomekLink, respectively, and jointly with different feature selection methods to preprocess the training set and select the best sampling method through comparative experiments, and the results are shown in **Table 3**.

(4) Feature Selection

After data cleaning and rebalancing, feature selection aims to extract the most informative variables that contribute significantly to classification. Data mining and machine learning rely heavily on feature selection, which involves reducing redundant and less correlated features to improve model performance [36]. Feature selection methods can be categorized into filter, wrapper, and embedded methods based on different evaluation techniques [37]. To determine the optimal preprocessing combination, we need to compare three representative algorithms of feature selection methods: mutual information classification (MIC), recursive feature elimination (RFE), and the Lasso algorithm, along with a combination of different sampling methods to preprocess the training set jointly. A comprehensive comparison shows that after the joint preprocessing of ADASYN and Lasso, the model has the highest accuracy, recall, F1 score, and AUC value, and the

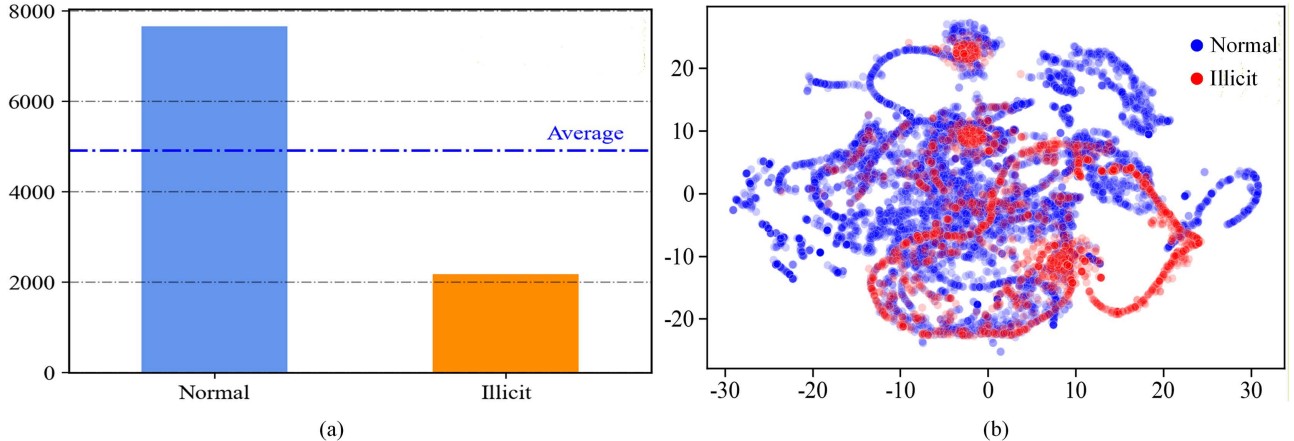

**Fig 2. Visualization plot of the dataset.**

**Table 3. Model performance comparison of joint preprocessing with sampling method and feature selection.**

| Sampling method | Category ratio | Feature Selection Method | Number of features selected | Acc/ % | Pre/ % | Rec/ % | F1/ % | AUC/ % |
|---|---|---|---|---|---|---|---|---|
| Raw data | 1975:6881 | Raw data | 812 | 99.49 | 99.50 | 98.04 | 98.77 | 98.96 |
| SMOTE | 6881:6881 | MIC | 50 | 99.70 | 100 | 98.53 | 99.26 | 99.26 |
| | | RFECV | 182 | 99.39 | 98.53 | 98.53 | 98.53 | 99.07 |
| | | Lasso | 61 | 99.39 | 99.01 | 98.04 | 98.52 | 98.89 |
| **ADASYN** | **6807:6881** | MIC | 50 | 99.49 | 99.50 | 98.04 | 98.77 | 98.96 |
| | | RFECV | 52 | 99.70 | 100 | 98.53 | 99.26 | 99.26 |
| | | **Lasso** | **71** | **99.70** | 99.51 | **99.02** | **99.26** | **99.45** |
| TomekLink | 1975:6663 | MIC | 50 | 99.49 | 100 | 97.55 | 98.76 | 98.77 |
| | | RFECV | 72 | 99.70 | 100 | 98.53 | 99.26 | 99.26 |
| | | Lasso | 50 | 99.49 | 99.50 | 98.04 | 98.77 | 98.96 |
| SMOTE+ TomekLink | 6745:6745 | MIC | 50 | 99.59 | 99.50 | 98.53 | 99.01 | 99.20 |
| | | RFECV | 82 | 99.49 | 98.54 | 99.02 | 98.78 | 99.32 |
| | | Lasso | 59 | 99.49 | 99.01 | 98.53 | 98.77 | 99.14 |

optimal number of features selected is 71. Therefore, in this study, the ADASYN oversampling algorithm and the Lasso feature selection method are selected for joint preprocessing of the dataset. As shown in **Fig 2(b)**, we use the t-SNE method to nonlinearly transform the original feature space to generate 2D t-SNE plots for the account category labels (normal or illicit) respectively. It can be seen that there are some distinguishable clusters in both categories, especially the red clustering chain near the bottom right region of the figure, and there are also 2 clustering points in the middle of the figure, which sufficiently emphasizes the necessity of applying machine learning techniques for classification.

### 4.3. Experimental comparison with existing work

To ensure a fair and rigorous performance comparison, this section evaluates the proposed ETHIAD model on the benchmark Ethereum transaction dataset introduced by Farrugia et al. [20], which has since become a de facto standard in Ethereum fraud detection research. Importantly, subsequent studies—LightGBM [21] and KNN [22]—retained the same dataset and data partitioning protocol, thereby providing a unified experimental foundation for direct and reproducible performance comparison. Accordingly, we benchmark ETHIAD against nine representative models from prior works, including transaction-feature-based approaches (XGBoost [20], LightGBM [21], KNN [22], DElightGBM [23], Adaboost [24], and Eth-PSD [25]) and graph-based methods, such as the graph embedding frameworks of Wu et al. [38] and Xia et al. [39], and the GNN-based PDGNN [40]. Model performance is comprehensively compared across accuracy, precision, recall, F1 score, and AUC (Table 4).

The results in Table 4 demonstrate that ETHIAD substantially outperforms all transaction-feature-based baselines trained on the same dataset. While earlier studies such as Farrugia et al. [20] (XGBoost, 96.30% accuracy), Ibrahim et al. [21] (LightGBM, 98.60%), and KNN [22] achieved strong performance using handcrafted transaction-level statistics, ETHIAD reaches 99.70% accuracy and 99.26% F1 score, highlighting its superior capacity to learn discriminative representations without manual feature engineering. Furthermore, compared with ensemble-based extensions such as DElightGBM [23], Adaboost [24], and Eth-PSD [25], ETHIAD consistently achieves higher precision and recall, demonstrating improved robustness in detecting illicit accounts.

When benchmarked against graph-structured models—including the embedding-based methods of Wu et al. [38] and Xia et al. [39], and the GNN-based PDGNN [40]—ETHIAD maintains a clear margin in accuracy and generalization while avoiding the scalability and complexity challenges inherent to graph construction and message passing. Collectively, these findings validate ETHIAD's effectiveness and its balanced integration of feature adaptability, interpretability, and computational efficiency.

From a computational perspective, ETHIAD offers notable efficiency advantages. The XGBoost algorithm scales approximately linearly with the number of samples and features ($O(n \times m \times \log k)$, where n is the number of samples, m is

**Table 4. Performance comparison with existing work.**

| Model | Acc/% | Pre/% | Rec/% | F1/% | AUC/% |
|---|---|---|---|---|---|
| XGboost [20] | 96.30 | / | / | 96.00 | 99.40 |
| LGBM [21] | 98.60 | 97.18 | / | 94.86 | / |
| KNN [22] | / | 98.60 | 98.80 | 98.70 | / |
| DElightGBM [23] | / | 81.96 | 80.50 | 81.22 | 80.97 |
| Adaboost [24] | / | 83.00 | 66.00 | 74.00 | 92.76 |
| Eth-PSD [25] | 98.11 | 98.00 | 98.00 | 98.00 | / |
| Wu et al. [38] | / | 92.70 | 89.30 | 90.80 | / |
| Xia et al. [39] | | 81.32 | 82.71 | 81.99 | |
| PDGNN [40] | 90.14 | 88.75 | 92.05 | 90.33 | |
| **ETHIAD** | **99.70** | **99.51** | **99.02** | **99.26** | **99.45** |

the number of features, and k is the maximum tree depth), whereas graph embedding and GNN methods require iterative computations over nodes and edges, leading to substantially higher time and memory consumption. Therefore, ETHIAD achieves a favorable balance of high predictive performance and computational efficiency, rendering it robust and practical for large-scale Ethereum illicit account detection.

## 4.4. Experimental comparison with mainstream models

Then, to further verify the effectiveness of the ETHIAD classification model, we compared ETHIAD with 10 mainstream machine learning algorithms, and the model parameters are set to default values. As **Table 5** shows, ETHIAD's overall performance surpasses other models. Tree-based ensemble models (such as RF, GBDT, LGBM, etc.) perform better than single models (such as MLP, SVM, LR, etc.) because they can capture complex nonlinear relationships between transaction data by combining multiple differentiated machine learning. The proposed model stands out among many ensemble models because ETHIAD's base model, XGboost, has significant advantages in processing large-scale, high-dimensional transaction data. It prevents model overfitting by introducing regularization terms and pruning strategies, improving the model's generalization ability and making its performance on unseen data more robust.

In **Fig 3**, we can compare the academic learning curves of ETHIAD and GBDT models. As the training sample size increases, ETHIAD's score on the training set always remains at 1, indicating that it still has overfitting. However, its overall fitting degree is significantly better than GBDT. On the other hand, GBDT has a relatively slow convergence speed, and the model's score on the validation set tends to be stable when the training sample size reaches 7,500, indicating underfitting. Meanwhile, ETHIAD's score on the validation set shows a rapid and continuous upward trend, suggesting the model may converge to better results with a further increase in the training sample size. Based on the existing data, ETHIAD outperforms GBDT regarding learning ability and fitting effect.

## 4.5. Hyperparametric optimization and generalization analysis

To further enhance the accuracy and robustness of ETHIAD, this study employs the Grid Search CV technique and F1-Score as the performance measure to determine the optimal parameter combination of the model. Five parameters are selected based on their significant impact on the model's performance: learning_rate, n_estimators, max_depth, subsample, and colsample_bytree. The default values for learning_rate and other parameters are initially set, and the 10-fold cross-validation method is utilized to determine the ideal number of base learners, n_estimators. Next, the other three main parameters are optimized after updating the adjusted parameters. Finally, by increasing (decreasing) the learning

Table 5. Performance comparison with mainstream machine learning models.

| Model | Acc/% | Pre/% | Rec/% | F1/% | AUC/% |
|---|---|---|---|---|---|
| MLP | 74.01 | 44.06 | 94.61 | 60.12 | 81.62 |
| SVM | 79.39 | 100.00 | 0.49 | 0.98 | 50.25 |
| KNN | 83.45 | 56.39 | 88.73 | 68.95 | 85.40 |
| Logistic Regression | 89.34 | 97.14 | 50.00 | 66.02 | 74.81 |
| Decision Tree | 99.09 | 97.56 | 98.04 | 97.80 | 98.70 |
| AdaBoost | 99.29 | 99.00 | 97.55 | 98.27 | 98.65 |
| GBDT | 99.29 | 97.58 | 99.02 | 98.30 | 99.19 |
| LGBM | 99.39 | 99.50 | 97.55 | 98.51 | 98.71 |
| Random Forest | 99.49 | 100.00 | 97.55 | 98.76 | 98.77 |
| Hist-GBDT | 99.49 | 99.50 | 98.04 | 98.77 | 98.96 |
| **ETHIAD** | 99.70 | 99.51 | 99.02 | 99.26 | 99.45 |

rate and simultaneously decreasing (increasing) the number of base learners, the optimal learning rate is determined when the model achieves the best score. As indicated in **Table 6**, the final parameters of learning_rate, n_estimators, max_depth, subsample, and colsample_bytree are 0.6, 140, 6, 1, and 1, respectively.

Validation curves for the learning_rate, n_estimators, and max_depth parameters are presented in **Fig 4** confirm the parameter tuning outcomes. The graphs illustrate that, with an increase in the learning_rate parameter, the model score

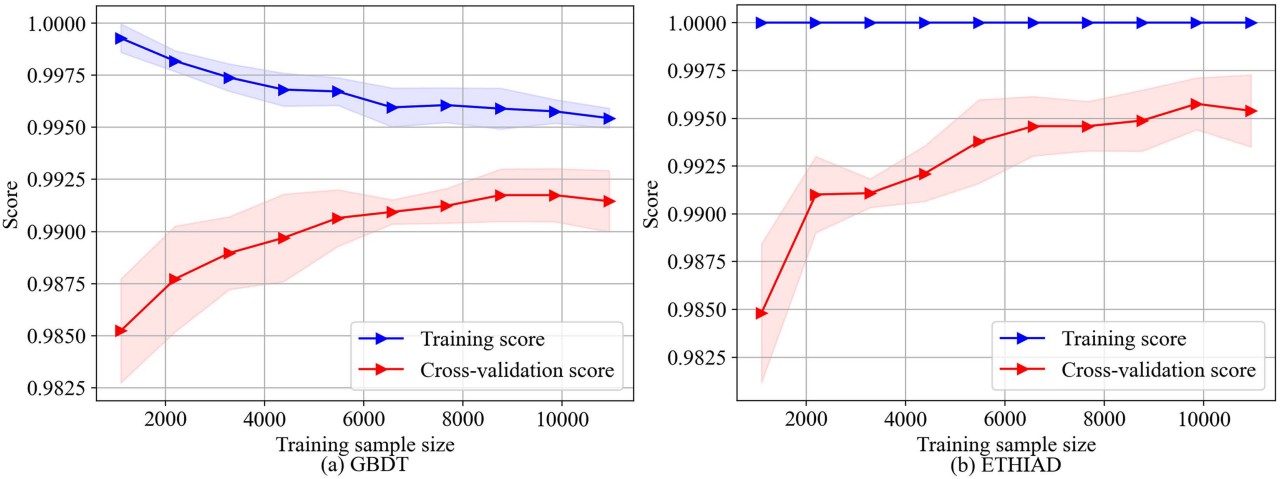

**Fig 3. Learning curve comparison between ETHIAD and GBDT.**

**Table 6. Hyperparameters explanation and tuning results.**

| Parameter name | Parameter explanation | Search Range | Tuning results |
|---|---|---|---|
| learning_rate | The weight of each newly added tree | [0.1, 1], step = 0.1 | 0.6 |
| n_estimators | Number of base learners | [60, 160], step = 10 | 140 |
| max_depth | Maximum depth of the tree | [1, 10], step = 1 | 6 |
| subsample | The proportion of random sampling in constructing each tree | [0.1, 1], step = 0.1 | 1 |
| colsample_bytree | The proportion of features randomly selected when constructing each tree | [0.1, 1], step = 0.1 | 1 |

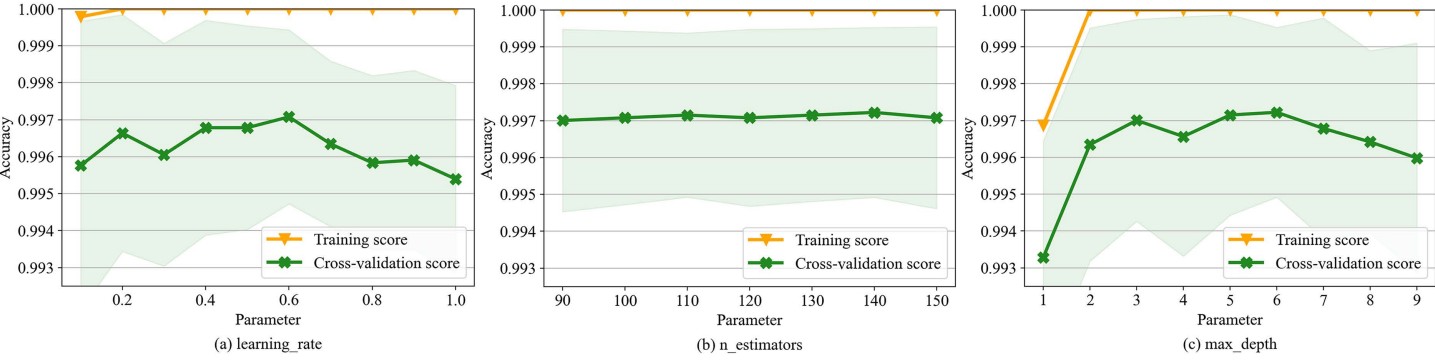

**Fig 4. Validation curves for the parameters learning_rate, n_estimators and max_depth.**

on the training set remains steady at 1.0. Furthermore, the model's score on the cross-validation set reaches its peak at a parameter value of 0.6, which indicates that the optimal value for this parameter is 120. Similarly, the best values for max_depth and max_features are 140 and 6, respectively.

As shown in **Table 7**, tuning the hyperparameters has significantly improved the model's overall performance. This implies that by systematically adjusting the hyperparameters, the model can better adapt to the features of Ethereum transaction data, thereby reducing the likelihood of overfitting.

To ensure that the ETHIAD model can accurately predict outcomes on new data, we used visualization tools such as the confusion matrix, KS curve, and ROC curve to evaluate the model's performance based on the best hyper-parameters on the test set. **Fig 5** compares the confusion matrix of ETHIAD with GBDT and Adaboost models on the test set, and it can be seen that the test set has a sample size of 985 and contains 781 normal accounts and 204 illicit accounts. The ETHIAD model correctly predicts all normal accounts and only incorrectly predicts one illicit account as a normal account. Nevertheless, ETHIAD still has the best prediction performance compared to other models.

In **Fig 6(a)**, the performance advantage of ETHIAD is clearly shown compared to the other models on the test set, our model has the largest AUC area. Meanwhile, **Fig 6(b)** displays the KS curve of ETHIAD, with the optimal classification threshold being 0.449 and a KS value of 0.995. **Fig 6(c)**, **(6d)** shows the cumulative gain curve and Lift curve of ETHIAD, respectively. The gain curve of the model indicates a positive offset, demonstrating that the classification ability of the two types of samples is significantly better than that of random selection. Overall, this analysis reveals that sample sampling and feature selection of Ethereum transaction data can effectively realize the feature modeling for the transaction network. This approach enables the model to capture essential information in the transaction data during training, enhancing its generalization ability in actual scenarios.

**Table 7. Model performance comparison before and after hyperparameter tuning.**

|  | Acc/% | Pre/% | Rec/% | F1/% | AUC/% |
|---|---|---|---|---|---|
| Before tuning | 99.70 | 99.51 | 99.02 | 99.26 | 99.45 |
| After tuning | 99.90 | 100 | 99.51 | 99.75 | 99.75 |

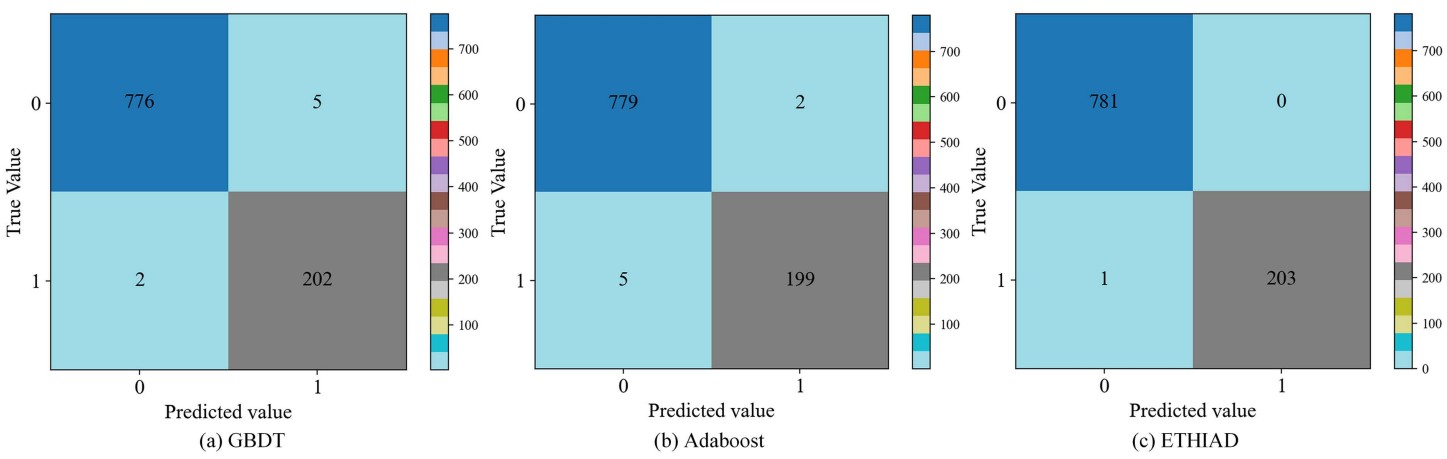

**Fig 5. Confusion matrix comparison of ETHIAD with other algorithms.**

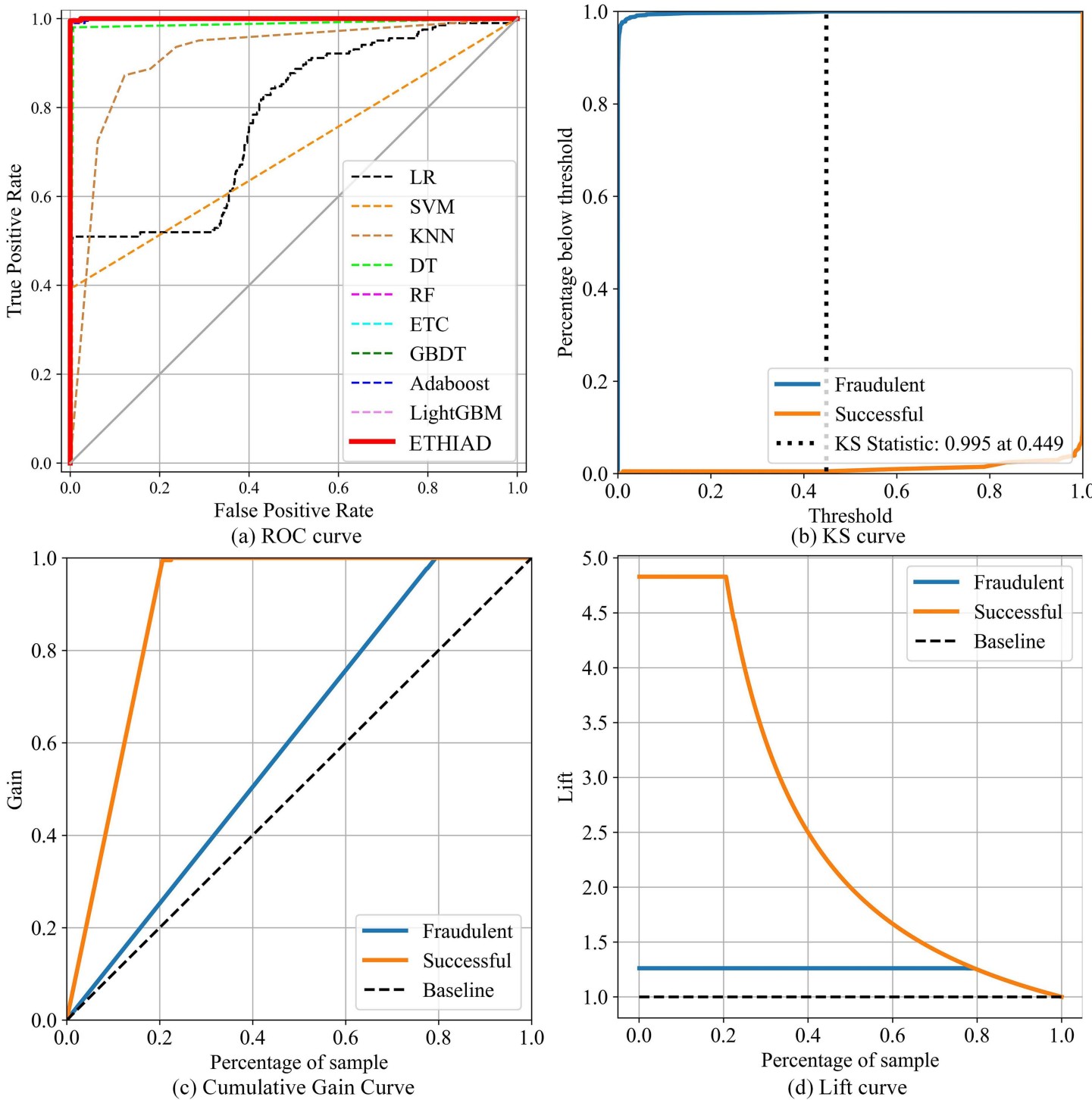

**Fig 6. Performance evaluation visualization of ETHIAD.**

## 5. Interpretability analysis of ETHIAD

To enhance the interpretability of the ETHIAD model, we employ the SHAP framework to analyze the key factors influencing illicit accounts on Ethereum from three perspectives: global feature importance, feature interaction, and individual sample decisions.

### 5.1. Analysis of the main factors influencing illicit accounts

**Fig 7** presents the feature importance ranking derived from four models. Although the ranking differs slightly among models, several key features consistently appear at the top, highlighting their critical role in identifying illicit accounts. For example, the feature *ERC20_most_sent_token_type_* ranks first in the Random Forest, XGBoost, and SHAP models, while *ERC20_total_ether_sent* ranks second in XGBoost and SHAP and fifth in LightGBM. Additional features such as *min_val_sent*, *Time_Diff_between_first_The_and_last_*, *ERC20_uniq_sent_addr*, and *Unique_Received_From_Addresses* also consistently rank highly, suggesting that they capture fundamental behavioral patterns of illicit accounts, including high token transfer activity, frequent interactions with multiple addresses, and irregular transaction timing—patterns commonly associated with phishing, laundering, and other fraudulent activities.

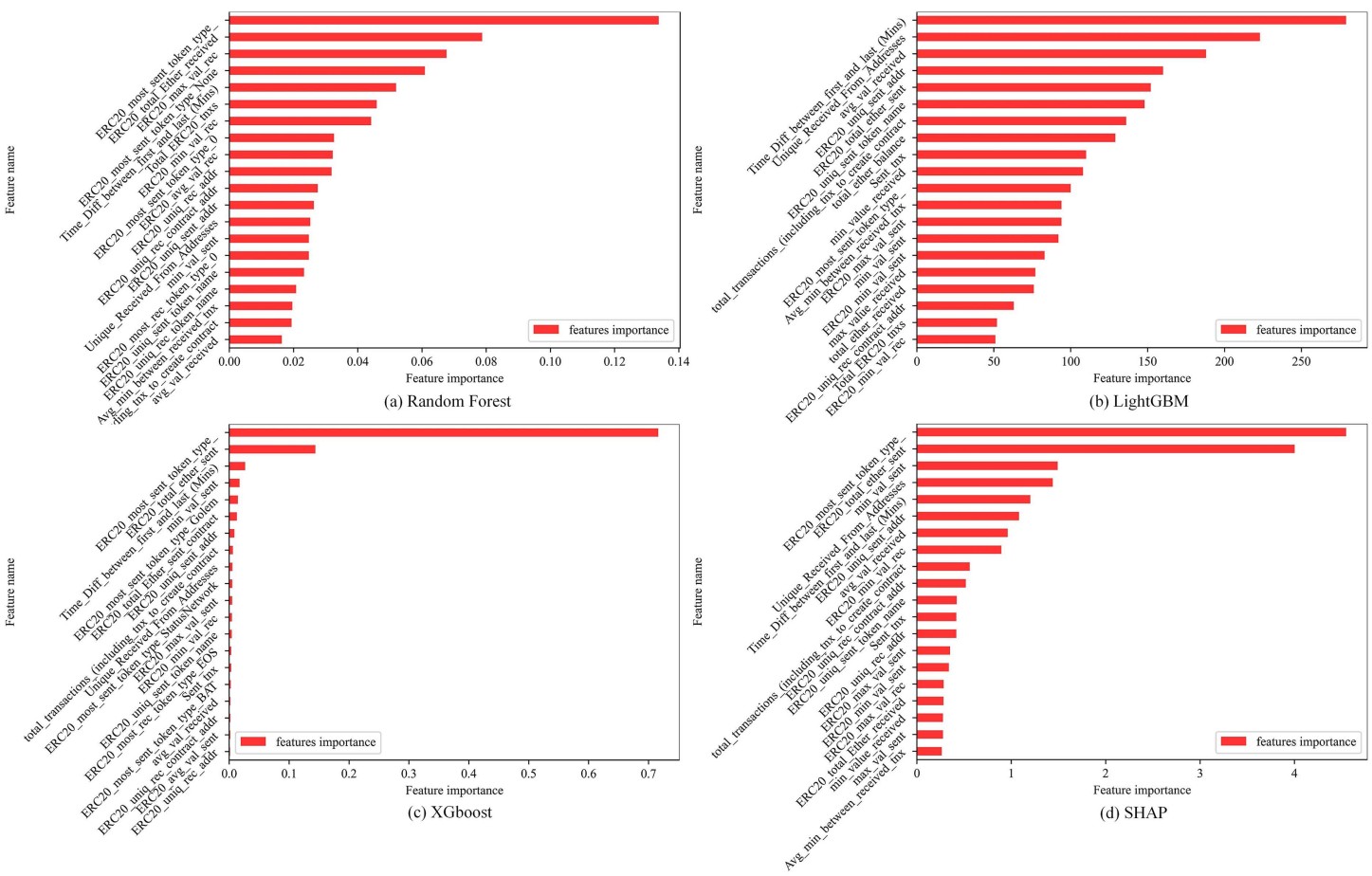

**Fig 7. Feature importance ranking of different models.**

The SHAP framework further quantifies the direction and magnitude of each feature's contribution. **Fig 8** displays the SHAP summary plot, where each point represents an account instance. The color gradient from red to blue corresponds to high-to-low feature values, and the horizontal position of each point indicates the feature's impact on the prediction. A positive SHAP value ($\phi(x_i^j) > 0$) indicates that the feature increases the likelihood of an account being classified as illicit, whereas a negative value ($\phi(x_i^j) < 0$) suggests a mitigating effect. This local interpretability allows analysts to understand why individual accounts are flagged, supporting targeted investigation and risk prioritization. Next, we analyze the key features identified by SHAP to explore their specific contributions and behavioral significance.

**(1) The type of tokens most sent via ERC20 transactions is the most critical factor in identifying illicit accounts**

The feature 'ERC20_most_sent_token_type_' is a binary feature encoded by One-Hot. It refers to the type of token that an Ethereum account sends the most via ERC20 transactions. The value of 1 indicates that the account does not have a token type. This feature is crucial in identifying illicit accounts. **Fig 8** shows that this feature contributes significantly to the model and positively affects the output of the results. The larger the sample value (i.e., 1), the larger the SHAP value, which drives the model to predict the sample as an illicit account. ERC-20 is a token standard on Ethereum that defines rules and conventions that make tokens issued on the Ethereum network interoperable with each other. Notably, tokens issued using the ERC-20 token standard are more comfortable to track and manage on Ethereum. Therefore, if an account sends tokens that do not comply with the ERC-20 standard, it may lead to a lack of transparency and standardization in transactions. This undoubtedly increases the potential risk of fraud.

**(2) The abnormally large total number of ERC20 token transactions sent is a distinctive feature of the illicit account.**

The feature 'ERC20_total_ether_sent' represents the total number of ERC20 token transactions an account sends. When analyzing the contribution of this feature to the model, it was found that its effect on the model is not clear when the sample value is small. However, it has a significant positive impact on the model when the sample value is abnormally

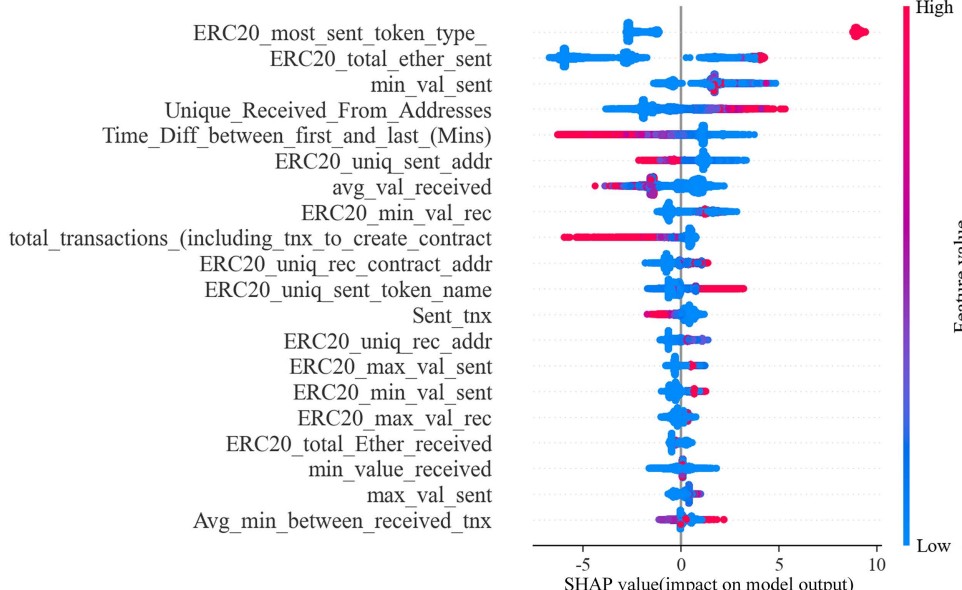

**Fig 8. SHAP summary plot of features.**

large. This means that an abnormally large transaction volume of ERC20 tokens sent by an account is a significant feature of an illicit account. The vast majority of Ethereum wallets and smart contracts follow the ERC-20 standard. As a result, normal business activities and investments may generate a large transaction volume in an account. However, accounts that engage in fraud, money laundering, and other illicit activities may have an unusually high transaction frequency and large batch transactions. Therefore, an abnormally large number of ERC20 token transactions sent is a distinguishing feature of an illicit account.

**(3)  It is important to note that when an account sends a large minimum amount of ETH, the risk of fraud in that account increases.**

The feature 'min_val_sent' represents the minimum value of ETH sent by the account. As depicted in **Fig 8**, the SHAP value of most of the samples of this feature is high, indicating that the positive and negative effects on the model are not significant. However, this feature has a strong interaction with other features, which makes its contribution to the model more significant. This interaction will be further analyzed in Section 5.2.

**(4)  Having a large number of unique addresses that a specific account receives transactions from can be a determining factor in identifying illicit accounts.**

The feature 'Unique_Received_From_Addresses' represents the total number of distinct addresses from which the account receives transactions. When the sample value of this feature is larger, the SHAP value is also larger, significantly affecting the model positively. Accounts involved in fraud, money laundering, or other illicit activities may receive large sums of money from multiple addresses or frequently conduct small transactions with many different addresses to evade tracking and monitoring. Therefore, having a high number of unique addresses from which an account receives transactions is essential in identifying illicit accounts.

**(5)  The distinguishing features of illicit accounts are frequent and large transactions that occur in a short period.**

The feature 'Time_Diff_between_first_and_last_ (Mins)' represents the time difference between the first and last transactions of the account. As shown in **Fig 8**, the smaller the value of this feature, the larger the SHAP value, which has a significant positive effect on the model. If an account has a large number of transactions in a short period, it could be a sign that the account is engaging in potentially illicit or high-risk financial activities, such as money laundering, illegal fundraising, or fraud. Therefore, frequent and large transactions in a short period are clear indicators of potentially illicit accounts.

In summary, the above analysis yields several of the most important factors that affect the model output. In addition, the features 'ERC20_uniq_sent_addr' (the number of ERC20 token transactions sent to the unique account address), 'avg_val_received' (the average value of Ether received), and 'total_transactions' (the total number of transactions on the account) have a significant negative effect on the model.

### 5.2.  Interaction analysis between important features influencing illicit accounts

The SHAP dependence plot in **Fig 9** illustrates the interactions between important features. The horizontal axis represents the sample value of the feature, while the vertical axis represents the corresponding SHAP value. The color bar from blue to red indicates the sample value of the interaction feature from small to large. As shown in **Fig 9(a)**, as the time difference between the account's first and last transaction increases, its SHAP value decreases, which has a negative effect on the model, indicating that accounts that make frequent transactions in a short period have a higher risk of fraud. According to the coloring of the interaction feature min_val_sent again, some of the accounts with small time difference of transactions send a higher minimum value of Ether, but these samples have a larger SHAP value, which drives the model to predict them as illicit accounts; as shown in **Fig 9(b)**, as the total

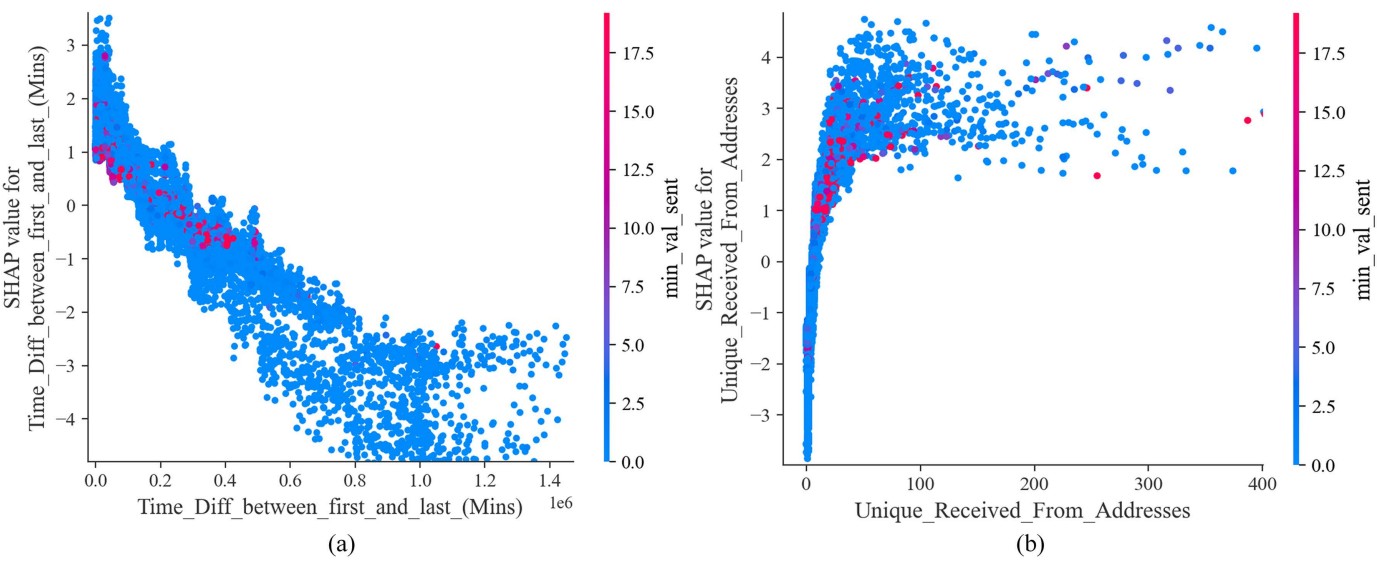

**Fig 9. SHAP dependence plot of features.**

number of unique addresses from which the account receives transactions increases, its SHAP value gradually increases, which has a positive effect on the model, indicating that accounts receiving transactions from multiple different addresses have a higher risk. According to the coloring of the interaction feature min_val_sent again, most of the accounts with larger min_values for sending Ether received transactions from multiple different accounts and these samples have larger SHAP values, driving the model to predict them as illicit accounts. Therefore, the above analysis shows that accounts that frequently make large transactions with multiple different addresses in a short period may have a high risk of fraud.

### 5.3. Sample decision analysis

In this section, we use the SHAP force plot, SHAP waterfall plot, and SHAP decision plot to visualize the prediction process of ETHIAD for a single sample and analyze the different roles of each feature in the sample decision process.

Fig 10 shows the prediction process of an account correctly predicted to be illicit. Fig 10(a) shows the SHAP force plot, $f(x)$ denotes the prediction value of the sample, the prediction process starts from the base value, the red arrow indicates that the feature pulls up the prediction value in the prediction process, the blue arrow indicates that the feature pulls down the prediction value, and the length of the arrow indicates the degree of contribution of the feature to the model prediction. From Fig 10(a), we can see the information of this account: there is no most sent type of ERC20 tokens, the transaction volume is 0, the time difference between the first and the last transaction is 49944.13 minutes (about 35 days), the total number of unique addresses that received the transaction is 26, and the minimum value of sent Ether is large, etc. This information indicates that the account has yet to send ERC20 tokens and frequently makes large transactions with multiple addresses in a short period. The model ultimately predicts the sample to be an illicit account. Fig 10(b),10(c) displays the SHAP waterfall plot and decision plot, showing each feature's positive and negative gains in the model's prediction process. The feature ERC20_most_sent_token_type_ has a value of 1 with the maximum positive gain, with a gain value of 8.9, while the feature ERC20_total_ether_sent has a value of 0 with the most significant negative gain and a gain value of −2.9.

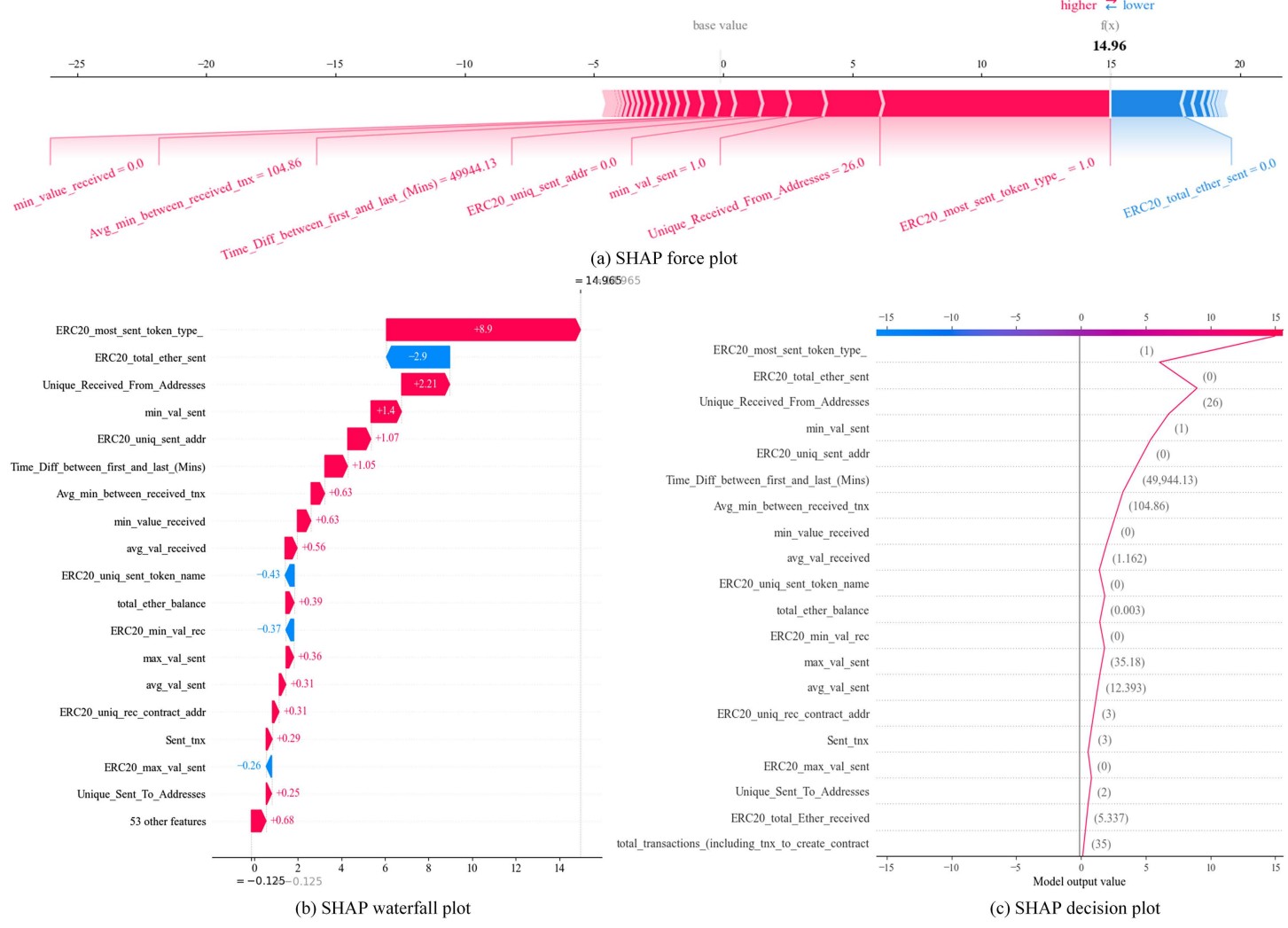

**Fig 10. Visualization of a case correctly predicted to be an illicit account.**

## 6. Discussion

In this study, we effectively implement the feature modeling for the Ethereum transaction network by sample sampling and feature selection of the basic transaction data of Ethereum and propose a novel explainable model for detecting illicit accounts on Ethereum, ETHIAD, based on the XGboost algorithm. In Section 4.3, we verify that ETHIAD outperforms existing models based on transactional data, and ETHIAD's comprehensiveness in terms of explainability is even more advantageous.

In terms of feature modeling for Ethereum transaction networks, the dataset used in this study is unbalanced, and One-Hot encoding of string features introduces sparsity and high dimensionality. To address these challenges, we applied ADASYN oversampling combined with Lasso feature selection, which not only balances the classes and reduces feature redundancy but also minimizes reliance on manual feature engineering. By systematically identifying the most informative features from high-dimensional transaction data, ETHIAD can automatically capture complex patterns of illicit activity. This

approach directly contributes to its superior performance over DElightGBM [23], enhancing both predictive accuracy and model robustness.

In terms of the explainability analysis of the model, we first identified the main factors affecting illicit accounts by calculating the importance of the feature using various models. We then focused on using the SHAP framework to analyze the contribution degree and positive and negative effects of some important features on the model output from multiple perspectives. Our conclusion suggests that accounts that frequently conduct large transactions with various addresses in a short period have a higher risk of fraud, which is consistent with the conclusion of Farrugia et al [20]. However, in contrast, the findings of this study indicate that accounts not transacted via ERC20 have a higher risk of fraud.

Accurate detection of illicit accounts will impact the illicit earnings of fraudsters, making ETHIAD models subject to well-designed AI adversarial attacks, such as Data Poisoning Attacks, Model Extraction Attacks, Model Inversion Attacks, etc. Additionally, changes in business rules, feature drift, label drift, or concept drift will likely impact the model's performance and usability over time. Therefore, in future research, we need to apply statistical or time-scale window methods to detect changes in data distribution at any time during the use of ETHIAD. We must also add adversarial samples and denoising methods to improve the model's ability to resist attacks when training a new model version.

## 7. Conclusions

This paper proposes a novel explainable illicit account detection model, ETHIAD, based on basic Ethereum transaction data. To achieve this, we first used the One-Hot encoding to mine the ERC20 token information of the account. We then address the dataset's category imbalance and feature sparse redundancy by using ADASYN oversampling and Lasso feature selection. Secondly, the XGboost algorithm is selected to train the illicit account detection model based on the preprocessed dataset, and the experiments show that the various performances of our model are improved by 0.05%−1.1% over the existing SOTA model, and it has a good generalization ability on unseen data. It also adequately validates that the feature engineering in this paper is more effective in achieving feature modeling for Ethereum transaction networks. Finally, we identify the main factors influencing illicit accounts in Ethereum by comparing the feature importance of multiple models and introduce the SHAP framework to analyze the contribution degree and the positive and negative effects of each feature on the model from multiple aspects, and the conclusion strongly enhances the explainability of ETHIAD.

## Supporting information

**S1 Appendix. Table A1. Complete list of the 42 extracted features.**
(DOCX)

## Author contributions

**Conceptualization:** Jiarong Lu, Bin Liao.

**Data curation:** Jiarong Lu, Bin Liao.

**Formal analysis:** Yi Liu, Kutorzi Edwin Yao.

**Funding acquisition:** Jiarong Lu.

**Methodology:** Jiarong Lu.

**Project administration:** Jiarong Lu.

**Resources:** Jiarong Lu, Bin Liao.

**Software:** Jiarong Lu, Bin Liao.

**Supervision:** Jiarong Lu, Bin Liao, Yi Liu, Kutorzi Edwin Yao.

**Validation:** Jiarong Lu, Bin Liao, Yi Liu, Kutorzi Edwin Yao.

**Visualization:** Jiarong Lu, Bin Liao, Yi Liu, Kutorzi Edwin Yao.

**Writing – original draft:** Jiarong Lu, Bin Liao, Yi Liu, Kutorzi Edwin Yao.

**Writing – review & editing:** Jiarong Lu, Bin Liao, Yi Liu, Kutorzi Edwin Yao.

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
