## [Decision Letter · Decision Letter 0]

27 Oct 2025

Dear Dr. Lu,

Thank you for submitting your manuscript to PLOS ONE. After careful consideration, we feel that it has merit but does not fully meet PLOS ONE’s publication criteria as it currently stands. Therefore, we invite you to submit a revised version of the manuscript that addresses the points raised during the review process.

Ensure all parameters used in all the equations are well defined and that the manuscript follows the submission guidelines of PLOS One.]

We look forward to receiving your revised manuscript.

Kind regards,

Ayei Egu Ibor, PhD

Academic Editor

PLOS ONE

Journal Requirements:

“This work was supported by the project from the Research Center for Social Economic Statistics and Big Data Applications in Xinjiang (XJEDU2024J100).”

6. Please note that your Data Availability Statement is currently missing the repository name and/or the DOI/accession number of each dataset OR a direct link to access each database. If your manuscript is accepted for publication, you will be asked to provide these details on a very short timeline. We therefore suggest that you provide this information now, though we will not hold up the peer review process if you are unable.

Reviewers' comments:

Reviewer's Responses to Questions

**Comments to the Author**

1. Is the manuscript technically sound, and do the data support the conclusions?

Reviewer #1: Partly

Reviewer #2: Yes

Reviewer #3: Yes

2. Has the statistical analysis been performed appropriately and rigorously?

Reviewer #1: I Don't Know

Reviewer #2: Yes

Reviewer #3: Yes

3. Have the authors made all data underlying the findings in their manuscript fully available?

Reviewer #1: Yes

Reviewer #2: Yes

Reviewer #3: No

4. Is the manuscript presented in an intelligible fashion and written in standard English?

Reviewer #1: Yes

Reviewer #2: Yes

Reviewer #3: Yes

Reviewer #1: Your manuscript presents a significant advancement in fraud detection within Ethereum networks. The use of ADASYN and Lasso for preprocessing and the SHAP framework for explainability are particularly commendable. I recommend discussing challenges encountered during model development and the implications of SHAP values more thoroughly.

Reviewer #2: 1. Is the manuscript technically sound, and do the data support the conclusions?

The manuscript is technically sound and presents a rigorous methodology for developing the ETHIAD model. However, it could be improved by explicitly stating the research objectives as questions or hypotheses. This would strengthen the scientific framework of the study and facilitate understanding for readers. The data used appear to align with the objectives, but a more thorough justification of the selected sources would enhance credibility.

2. Has the statistical analysis been performed appropriately and rigorously?

Yes, the use of the XGBoost algorithm combined with SHAP for explainability is well-justified and methodologically sound. However, a clearer transition between feature modeling and explanatory analysis, as well as clarification of the precise role of each step in the algorithm, would be beneficial for readers less familiar with these techniques.

3. Have the authors made all data underlying the findings in their manuscript fully available?

The sources of the variables are mentioned, but a detailed justification of the data selection criteria is missing. Providing an explanation of the relevance and reliability of the data would enhance the transparency and reproducibility of the results.

4. Is the manuscript presented in an intelligible fashion and written in standard English?

The manuscript is well-structured and written in standard English. However, a clearer articulation of the contributions as research objectives or questions right from the introduction would improve readability and the overall impact of the text.

Reviewer #3: The paper explores an intriguing research area by introducing a novel explainable model for detecting illicit accounts on Ethereum, called ETHIAD. Additionally, the manuscript employs the SHAP framework to enhance the explainability of its findings. However, there are several areas where the manuscript could be improved:

- The provided dataset link (https://github.com/Lujiarong1203/ETHIAD) is currently invalid, which makes it difficult to verify several of the claims made in the manuscript. It is important to ensure that the dataset is accessible, either through a valid public link or by providing an alternative means of access, to support the transparency and reproducibility of the study.

- The research contribution section should be refined to avoid overclaiming. For instance, the statement “we introduce the SHAP framework” could be misleading, as it implies that the authors developed the SHAP framework themselves. A more accurate phrasing such as “we use the SHAP framework” or “we apply the SHAP framework in our analysis” would better reflect the authors’ actual contribution and maintain clarity in presentation.

- The statement “the transparent and open nature of the blockchain's transaction information makes it easy to track transactions” requires further clarification. The authors should elaborate on what specific features of blockchain transparency facilitate transaction tracking and explain how this property is leveraged within their proposed model or analysis.

- In addition, the manuscript would benefit from a discussion on the computational complexity of the proposed model compared to state-of-the-art (SOTA) approaches. Providing such a comparison would help readers understand the efficiency of the proposed method. It is also important to elaborate on how the model reduces the dependence on manual feature design relative to existing solutions, as this is a key aspect of innovation and practicality.

- The reference to Farrugia et al. [20] requires further detail. The manuscript does not clearly describe the methods used in their fraud detection model—whether they were based on basic transaction data techniques or more advanced approaches. Including this information would provide better context and strengthen the comparative analysis.

- Furthermore, it is unclear whether the proposed method was tested on the same dataset as those used by the SOTA solutions. Clarifying this point would enhance the credibility of the performance comparison and help readers evaluate the relative effectiveness of the proposed approach.

- The authors claim that existing datasets suffer from category imbalance, which distorts model performance. However, no supporting evidence or references are provided to substantiate this statement. Including empirical data or literature support would significantly strengthen this argument.

-Also, the selection of algorithms—Random Forest, LightGBM, and XGBoost—requires further justification. The authors should explain why other commonly used algorithms such as Decision Trees, ETH-PSD, AdaBoost, or KNN were not considered, particularly since they have been utilized in related studies. Including a brief comparison or rationale would help demonstrate the robustness and relevance of the chosen methods.

- Additionally, several figures referenced in the text appear to be missing from the manuscript. The authors should ensure that all figures are included, properly labeled, and clearly referenced within the text to support the narrative.

**Do you want your identity to be public for this peer review?** For information about this choice, including consent withdrawal, please see our Privacy Policy

Reviewer #1: **Yes: ** Halilibrahim Gökgöz

Reviewer #2: No

Reviewer #3: No

---

## [Author Response · Author response to Decision Letter 1]

7 Nov 2025

The authors sincerely appreciate the constructive comments provided by the reviewers and the editor, and have carefully addressed all suggestions. Point-by-point responses are provided in the “Response to Reviewers,” and all corresponding revisions are highlighted in red in the “Revised Manuscript with Track Changes.”

---

## [Decision Letter · Decision Letter 1]

24 Nov 2025

ETHIAD: A novel explainable model for detecting illicit accounts on Ethereum

PONE-D-24-49589R1

Dear Author,

We’re pleased to inform you that your manuscript has been judged scientifically suitable for publication and will be formally accepted for publication once it meets all outstanding technical requirements.

Kind regards,

Ayei Egu Ibor, PhD

Academic Editor

PLOS ONE

Additional Editor Comments (optional):

Reviewers' comments:

Reviewer's Responses to Questions

**Comments to the Author**

Reviewer #3: All comments have been addressed

2. Is the manuscript technically sound, and do the data support the conclusions?

Reviewer #3: Yes

3. Has the statistical analysis been performed appropriately and rigorously?

Reviewer #3: Yes

4. Have the authors made all data underlying the findings in their manuscript fully available?

Reviewer #3: Yes

5. Is the manuscript presented in an intelligible fashion and written in standard English?

Reviewer #3: Yes

Reviewer #3: The revised manuscript is now ready for publication. The authors have addressed all previous comments, which have significantly enhanced the clarity and quality of the work. However, before the final submission, please ensure that all citations are placed before the period. I have highlighted this issue on lines 104 and 106 as examples. While this is a minor formatting detail, it is important for maintaining consistency and adhering to publication guidelines.

**Do you want your identity to be public for this peer review?** For information about this choice, including consent withdrawal, please see our Privacy Policy

Reviewer #3: No

---

## [Editor Report · Acceptance letter]

PONE-D-24-49589R1

PLOS ONE

Dear Dr. Lu,

I'm pleased to inform you that your manuscript has been deemed suitable for publication in PLOS ONE. Congratulations! Your manuscript is now being handed over to our production team.

Kind regards,

on behalf of

Dr. Ayei Egu Ibor

Academic Editor

PLOS ONE